# The Interplay of Personality Traits and Psychological Well-Being: Insights from a Study of Italian Undergraduates

**DOI:** 10.3390/ijerph22020132

**Published:** 2025-01-21

**Authors:** Graziella Orrù, Andrea Piarulli, Ciro Conversano, Giovanna Grenno, Angelo Gemignani

**Affiliations:** Department of Surgical, Medical, Molecular & Critical Area Pathology, University of Pisa, via Savi, 10, 56126 Pisa, Italy; andrea.piarulli@unipi.it (A.P.); ciro.conversano@unipi.it (C.C.); giovanna.grenno@med.unipi.it (G.G.); angelo.gemignani@unipi.it (A.G.)

**Keywords:** Big Five Inventory, BFI, mental health, personality traits, psychological well-being

## Abstract

Objectives: The mental health and well-being of university students are crucial areas of research due to their significant impact on academic success, personal development, and overall life satisfaction. Since previous research consistently shows that the stress related to academic challenges can significantly affect mental health, the study aims to examine the relationship between personality traits, locus of control, and psychological well-being outcomes. Methods: In the present study, 67 Italian undergraduate students (19.45 ± 1.62 years) were evaluated to assess the connections between personality traits, locus of control, and key psychological outcomes including depression, anxiety, sleep disturbances, and subjective memory complaints. Results: The analyses of the collected variables revealed a strong interrelationship between stress, anxiety, depression, and insomnia, contributing to a general construct termed psychological well-being disturbances (PWBD). When considering the Big Five personality traits, a significant positive association with internal control and mindfulness levels was observed both for extraversion and conscientiousness, while neuroticism was associated with increased external control and subjective memory complaints. Conclusions: These findings highlight the importance of understanding personality traits in shaping psychological well-being and coping mechanisms among students. Educational institutions should consider incorporating personality-based approaches into their support systems, as fostering traits linked to better psychological well-being, such as extraversion, conscientiousness, and agreeableness, may help mitigate the negative impact of stress and anxiety commonly experienced in academic settings.

## 1. Introduction

The mental health and well-being of university students have become a critical focus of research due to their profound implications for academic success, personal development, and overall life satisfaction. The process of transitioning from childhood to adulthood, typically characterized by the commencement of college, represents a difficult and stressful period; in fact, this phase is marked by increased academic pressures, novel social challenges, and greater responsibility for one’s life decisions, all of which can significantly impact mental health. Previous research (i.e., [1,2]) has consistently demonstrated that students who report elevated levels of stress, anxiety, depression, and even suicidal ideation are at heightened risk of experiencing both mental health difficulties and academic struggles ([3]; for a meta-analysis, see: [4]).

Understanding psychological factors influencing student well-being [5] is crucial for identifying at-risk individuals, enabling early detection, and developing targeted interventions. Mindfulness abilities, for example, have been shown to play a crucial role in managing stress and improving mental health outcomes (i.e., [6,7]). Evidence supporting mindfulness-based interventions (MBIs) shows significant benefits in reducing stress, anxiety, and depression by enhancing self-awareness and emotional regulation, thereby improving overall mental well-being (i.e., [6]).

A key psychological construct closely associated with mental health outcomes and overall well-being state is the locus of control (LOC) [8]. It refers to the extent to which individuals believe they have control over the events in their lives, with their control being either primarily internal or external. LOC has been shown to significantly influence students’ ability to manage stress and cope with different mental health challenges [9]. Internal control, the belief that one’s actions can influence outcomes, has been associated with better mental health outcomes (i.e., [10]) and academic performance (i.e., [11]), while external control, the belief that outcomes are determined by external factors such as fate, luck, or the influence of others, has been linked to higher levels of stress and anxiety [12].

Stress is another critical factor in the mental health landscape of university students. Chronic stress can lead to a wide range of negative psychological outcomes, including depression, anxiety, and sleep disturbances (i.e., [13]). Indeed, depression and anxiety represent two of the most common mental health issues among students. A systematic review from Ibrahim and colleagues [14] revealed that prevalence rates of depression among university students ranged from 10% to 85%, with an average of 30.6%. Similarly, other studies observed a rise in mental health issues, such as anxiety and depression, and perceived stress among university students [15]. Given their high prevalence and potentially debilitating effects, exploring the relationships between depression, anxiety, and other psychological factors is crucial. Sleep disturbances are another significant concern [16]. In this context, Lund et al., [17] found that over 60% of university students were classified as poor-quality sleepers, and Gaultney [16] reported 27% were at risk for at least one sleep disorder. Sleep disturbances not only lead to exhaustion and poor cognitive performance but also exacerbate pre-existing mental health issues (i.e., [13]).

Memory difficulties are another common issue among students, including subjective memory complaints (SMC); SMC refers to subjective memory loss or subjective memory impairment [18] and can significantly impact students’ confidence in their academic abilities and daily functioning, whether real or perceived.

A growing attention has also been given to the role of personality traits, as outlined by the Big Five model [19,20]: extraversion, agreeableness, conscientiousness, neuroticism, and openness, in shaping mental health outcomes in students. For instance, a higher level of neuroticism has consistently been associated with increased susceptibility to anxiety, depression, and stress [21]. Conversely, students with higher levels of extraversion and agreeableness tend to present more positive mental health outcomes, including better emotional regulation and academic resilience [22].

The primary aim of this study was to investigate how personality traits, specifically the Big Five dimensions, interacted with locus of control (LOC) to influence various psychological outcomes among first-year university students. We hypothesized that personality traits, particularly extraversion, conscientiousness, and neuroticism, would have a significant impact on psychological well-being, with internal LOC potentially serving as a moderating factor. Specifically, we expected that students with higher levels of extraversion and conscientiousness would experience better psychological outcomes, while those exhibiting higher neuroticism and external LOC would report more negative psychological effects, including increased levels of anxiety, depression, and sleep disturbances. This research aimed to provide insights that could guide the development of targeted interventions to support the mental health and well-being of university students, particularly in their first year, and enhance their academic success.

## 2. Methods and Measures

### 2.1. Participants and Procedure

Sixty-seven Italian undergraduate students (59 females) voluntarily participated in a comprehensive evaluation, completing validated questionnaires to investigate various psychological features.

Data collection was conducted via an online survey platform (Google Form). Participants were informed that the purpose of the research was to psychometrically assess personality domains and their association with indicators of mental well-being through the administration of specific questionnaires. General instructions provided to the participants in the survey were as follows: ‘*We are conducting research aimed at psychometrically assessing personality domains and their association with indicators of mental well-being by administering specific questionnaires/tests. Completing the questionnaire will take approximately 15 minutes*’. For each individual questionnaire, participants were provided with the specific instructions relevant to that questionnaire, which remained consistent with the original instructions for each test.

The cohort of volunteers had a mean age of 19.45 ± 1.62 years (mean ± standard deviation, SD), and a mean education level of 13.04 ± 0.37.

All participants provided informed consent before starting the online survey, which was distributed via a link in January 2024. The experimental protocol (#0011991/2023) was approved by the Bioethics Committee of the University of Pisa and adhered to the guidelines of the Declaration of Helsinki and its later amendments. For each volunteer, age, gender, and education level (in years) information was collected.

### 2.2. Psychometric Measures

The online survey comprised the following list of questionnaires:

**Locus of controls Behavior (LCB)** [23,24]: LCB is a questionnaire consisting of 17 questions with a rating scale from 0 to 5 measuring an individual’s “locus of control” (i.e., internal: “*I can anticipate difficulties and take action to avoid them*”; external: “*a great deal of what happens to me is probably just a matter of chance*”). The scale demonstrated good reliability, with Cronbach’s Alpha values of 0.70 and 0.73 for internal locus and external locus, respectively [24].

**Beck Depression Inventory-II (BDI-II)** [25,26]: BDI-II is a 21-item questionnaire assessing the intensity of depressive symptoms in adults and adolescents [25,27]. The respondent is asked to reflect on the past two weeks and rate the relevance of each statement (i.e., “*I am much more irritable than usual*” or “ *I am too tired or fatigued to do most of the things I used to do*”) related to the following symptoms: sadness, pessimism, sense of failure, loss of pleasure, guilt, expectation of punishment, dislike of self, self-accusation, suicidal ideation, episodes of crying, irritability, social withdrawal, indecisiveness, worthlessness, loss of energy, insomnia, irritability, loss of appetite, preoccupation, fatigue, and diminished interest in sexual activity [28]. The BDI-II has demonstrated strong psychometric properties, including good internal consistency, reliable test-retest stability, and construct validity [26].

**State-Trait Anxiety Inventory (STAI-Y*)*** [29,30]: The inventory comprises 40 self-report items rated on a 4-point Likert scale. It is divided into two subscales: STAI-Y1 assesses state anxiety, while STAI-Y2 evaluates trait anxiety. State anxiety includes items such as: “*I am tense*; “*I feel upset*”; and “*I am relaxed*; “*I feel secure*”. Trait anxiety items include: “*I worry too much over something that really doesn’t matter*”; “*I make decisions easily*”; “*I am a steady person*” [31]. The instrument is a suitable and reliable tool for assessing anxiety in both research and clinical settings [32].

**Big Five Inventory (BFI)** [22,33,34]: The BFI consists of 44 statements (i.e., “*Does things carefully and completely*” or “*Worries a lot*”) rated on a 5-Likert scale and it assesses extraversion, agreeableness, conscientiousness, neuroticism, and openness. The self-report scale provides a concise and prototypical measure of core personality traits, such as extraversion, agreeableness, conscientiousness, neuroticism, and openness to experience [22,33,35].

**Perceived Stress Scale (PSS)** [36,37]: The PSS is one of the most used psychological tools for assessing the subjective level of stress, which measures the extent to which individuals evaluate situations in their lives as stressful and consists of 10 items (i.e., “*In the last month, how often have you felt nervous and stressed?*”; “*In the last month, how often have you been able to control irritations in your life?*”). Higher scores reflect a greater tendency to evaluate situations as stressful. PSS total scores can range from 0 to 40 and cut-offs are as follows: (i) 0–13 low stress, (ii) 14–26 moderate stress, and (iii) 27–40 high perceived stress.

**Prospective and Retrospective Memory Questionnaire (PRMQ)** [38,39]: The PRMQ is a self-reported psychometric questionnaire with 16 items rated on a 5-point Likert and assesses common everyday memory lapses for both prospective (PRMQ-P) (i.e., “*Do you decide to do something in a few minutes’ time and then forget to do it*”) and retrospective (PRMQ-R) (i.e., “*Do you fail to recognize a place you have visited before*”) memory.

**Mindfulness Attention Awareness Scale** (**MAAS)** [40,41]: The MAAS is a 6-point Likert scale questionnaire consisting of 15 items (i.e., “*I rush through activities without being really attentive to them*”; “*I do jobs or tasks automatically, without being aware of what I’m doing*”), designed to assess dispositional mindfulness. Dispositional mindfulness refers to an individual’s tendency to remain in a mindful state over time, through receptive awareness and attention to what is occurring in the present moment [40]. The Italian adaptation of the MAAS exhibits good psychometric properties, with its scores showing no influence from participants’ gender [42].

**Ford Insomnia Response to Stress Test (FIRST)** [43,44,45]: The FIRST is a 9-item self-assessment tool designed to identify individuals at risk of developing insomnia due to stressful life events (i.e., “*How likely are you to have trouble falling asleep the night before an important meeting or presentation?*”; “*How likely are you to have trouble falling asleep the night before an important meeting the next day?*”). It evaluates the psychophysiological hyperarousal that individuals may experience in stressful situations, which subsequently disrupts their ability to achieve restful sleep [45].

**Insomnia Severity Index** (**ISI)** [46,47]: The ISI is a brief, 5-item self-report questionnaire developed to assess the nature and severity of insomnia over the past two weeks (i.e., “*How worried/distressed are you about your current sleep problem*” or “*How noticeable to others do you think your sleeping problem is in terms of impairing the quality of your life?*”). It is one of the most widely used tools for assessing insomnia severity, examining aspects such as sleep onset time, sleep continuity, wakefulness, sleep satisfaction, and the impact of insomnia on daily life and interpersonal relationships [48]. The ISI questionnaire has good psychometric properties and appears to be reliable for detecting insomnia severity and identifying patients’ symptoms [47].

### 2.3. Data Analysis

#### 2.3.1. Statistical Methods Overview

We present in detail the statistical methods used in the following section. For each demographic and psychometric variable (except for gender), both mean ± standard error and median and interquartile range were estimated. The reliability of each psychometric scale was assessed by calculating its Cronbach’s alpha value.

Note that for all the analyses, the threshold for significance was set at *p* = 0.05.

#### 2.3.2. Correlations

The degree of association between specific pairs of variables was estimated using the Pearson correlation coefficient. The r-value significance was assessed by conducting a permutation test on the r-statistics based on 1000 randomization [49]. This procedure was chosen as permutation tests are robust to violations of parametric statistics assumptions such as normality and heteroscedasticity and are thus well suited for the analyses of psychometric questionnaires’ scores.

#### 2.3.3. Principal Components Analysis

The reliability (and hence the statistical power) of a PCA depends on two sufficient conditions (see [50,51,52]):A sample size (i.e., number of observations/participants) higher than 50.A sample size-to-variable ratio ≥ 5.

Throughout the years, the determination of the minimum sample size and the sample size-to-variable ratio has been a matter of ongoing debate among researchers: it is thus fair to underline that, as demonstrated by de Winter and colleagues [52], when data are well conditioned (high loadings, low number of factors, relatively high number of variables), the two conditions can be relaxed (e.g., a PCA can yield reliable results also for sample sizes well below 50).

However, to ensure the reliability of our findings, we adopted the two thresholds mentioned at the beginning of the paragraph. Indeed, each PCA conducted in the study satisfied the two conditions: the sample size was always 67, and the sample size-to-variable ratio was higher than 10 (6 variables for each PCA), and thus well above the lower limit of 5. The suitability of a dataset for a PCA was verified using Kaiser–Meyer–Olkin measure of sampling adequacy [53] and Bartlett’s test of sphericity [54].

The measure of sampling adequacy is a statistic that indicates the proportion of variance in the dataset that might be caused by common underlying factors. Values higher than 0.50 indicate that the dataset’s variables share an adequate level of variance with each other, hence the appropriateness and reliability of a PCA.

Bartlett’s test verifies whether the correlation matrix estimated on the dataset’s variables is significantly different from an identity matrix. *p*-values lower than 0.05 indicate the existence of correlations between variables and hence the suitability of the dataset for a structure detection procedure (i.e., PCA).

For each PCA, the number of retained components was determined using Horn’s parallel analysis ([55], see Appendix A). For each retained component, the variables’ loadings were extracted. Note that the component’s loadings are the Pearson’s correlation coefficients between each variable and the component itself: when two or more variables show significant loadings on the same component, this suggests the existence of a common underlying factor contributing to the variables’ “behavior”. For each component, the loadings’ significance was estimated using a single-threshold test for the maximum r-statistics ([56], see Appendix A), thus addressing the multiple testing issue (i.e., simultaneous testing on multiple correlations). The single threshold test for the maximum r-statistics was chosen since (i) it does not require any assumption about data distribution (e.g., normality) and (ii) it is a robust approach to control for type I statistical errors (i.e., rejection of a true null hypothesis).

All statistical analyses were conducted using tailored codes written in MATLAB (R2021b, MathWorks, Natick, MA, USA).

## 3. Results

### 3.1. Descriptives

Sixty-seven Italian undergraduate students (first year) enrolled in a psychology degree program voluntarily underwent a comprehensive psychometric evaluation.

For each variable (demographic and psychometric), the population mean, standard deviation, median, and interquartile range are reported in Table 1.

Participants exhibited subjectively perceived stress levels ranging from moderate to high, with a mean score of 26.81 (SD = 6.11), a median of 26, and an IQR of 7.50 on the PSS. The level of insomnia was subthreshold, with a mean score of 8.43 (SD = 4.97), a median of 8, and an IQR of 7.50 on the ISI. Additionally, participants reported mild depression with a mean score of 17.91 (SD = 9.75), a median of 18, and an IQR of 15.25 on the BDI-II, as well as mild anxiety, with a mean score of 45.12 (SD = 11.45), a median of 44, and an IQR of 16.50 on the STAY-Y1. All psychometric scales showed satisfactory internal consistency levels (see Cronbach’s Alpha, Table 1): all the scales except for LCB-I (0.681) had alpha values higher than 0.7, although values higher than 0.6 are still considered adequate [57]. As apparent from Table 1, both age and education level showed an extremely low variance level and, therefore, were not included in the following PCA analyses.

### 3.2. Stress, Anxiety, Depression, and Insomnia Share a Common Underlying Factor

We first examined the dataset composed of PSS, STAY-Y1, STAI-Y2, BDI-II, ISI, and FIRST. The dataset satisfied both Kaiser–Meyer–Olkin measure of sampling adequacy and Bartlett’s test (0.84 and *p* < 0.001, respectively) and was thus submitted to a PCA. One component, accounting for 58.5% of the total variance, was extracted (see Appendix A).

All psychometric questionnaires showed highly significant loadings on the selected PCA component (*p* < 0.001 for all loadings, see Figure 1 and Table 2 below).

This component reliably “explained” all the considered variables; thus, it was used as a single factor incorporating all the psychological well-being domains. We will refer to the factor, as anticipated in the Section 2, as PWBD. Note that high PWBD levels correspond to a low overall psychological well-being, while lower levels reflect a higher psychological well-being.

### 3.3. High Association Between Prospective and Retrospective Memory Impairment

A significantly high correlation was observed between PRMQ-P and PRMQ-R subscales (r = 0.63, *p* < 0.001). Consequently, only the PMRQ total score was used for subsequent analysis.

### 3.4. Dissociation Between Internal and External Locus of Control Behavior

An examination of the relationships between internal and external LOC behavior revealed a significant anti-correlated relationship between the two subscales (r = −0.43, *p* < 0.001). Both subscales were therefore retained for further analysis. 

### 3.5. Big Five Personality Traits

We finally considered the Big Five Inventory personality traits: extraversion, agreeableness, conscientiousness, neuroticism, and openness. We explored the relationship between each trait and the following factors (selected based on previous analysis outcomes, see Section 3.2, Section 3.3 and Section 3.4):-Psychological well-being disturbances (PWBD)-Total perceived memory impairment (PRMQ)-Internal locus of control behavior (LCB-I)-External locus of control behavior (LCB-E)-Mindful Attention Awareness Scale (MAAS)

For each personality trait, we thus performed a PCA including both the trait to be investigated and the selected factors.

#### 3.5.1. Extraversion

One component accounting for 48% of the total variance was extracted (KMO = 0.640, Bartlett’s test *p* < 0.001, Appendix A). All variables had significant loadings on the component: extraversion, internal LOC, and MAAS showed positive loadings, whereas external LOC, memory complaints (PRMQ), and PWBD showed negative ones (see Table 3 and Figure 2). High extraversion is thus paralleled by a higher internal control and higher mindfulness, attention, and awareness (MAAS) and, at the same time, by a lower external control, lower memory complaints, and lower PWBD levels (i.e., higher psychological well-being).

#### 3.5.2. Agreeableness

Two PCA components were extracted (KMO = 0.690, Bartlett’s *p* < 0.001), the first component accounting for 47% of the total variance, and the second for 23% (Appendix A).

Agreeableness showed a highly significant positive loading on the second component together with MAAS, while both PRMQ and PWBD had significant negative loadings (see Table 4, Figure 2, and Appendix A). Neither external nor internal control had a relationship with the component.

Subjects with a higher agreeableness trait showed higher mindfulness levels along with lower memory complaints and lower PWBD (i.e., lower levels of stress, anxiety, depression, and insomnia), independently from the LOC.

#### 3.5.3. Conscientiousness

We next considered the conscientiousness trait. The dataset had a KMO score of 0.67 and Bartlett’s *p* lower than 0.001. One component (48% of the total variance) was extracted (Appendix A). All variables had significant loadings on the retained component: conscientiousness along with internal LOC and MAAS showed positive loadings, whereas external LOC, PRMQ, and PWBD showed negative ones (Table 5 and Figure 2).

Individuals with a high conscientiousness trait were thus characterized by a higher internal control and mindfulness while having lower external control, memory complaints, and PWBD.

#### 3.5.4. Neuroticism

One component was retained (KMO = 0.704, Bartlett’s *p* < 0.001), which accounted for 49% of the total dataset’s variance (Appendix A). All variables had significant loadings on the retained component (Table 6 and Figure 2).

Subjects with higher neuroticism traits had higher external control, memory complaints, and PWBD. At the same time, they showed a lower internal control, paralleled by a lower mindfulness.

#### 3.5.5. Openness

Finally, we considered the openness trait. The dataset satisfied both KMO and Bartlett’s test of sphericity (0.640 and *p* < 0.001, respectively). Two components were retained, respectively accounting for 46.5% and 24.5% of the variance (Appendix A).

Openness had an extremely significant loading on the second PCA component, together with internal LOC. No other variable had any significant loading on the considered component (see Table 7, Figure 2, and Appendix A).

Thus, at least for our dataset, subjects characterized by a high openness were those showing a higher internal control. None of the other psychometric variables had a relationship with this particular trait.

## 4. Discussion

The results of the present study provide significant insights into the relationships between personality traits and psychological well-being domains among a sample of Italian students. The psychometric assessment of the participants revealed a strong interrelationship among stress, anxiety, depression, and insomnia, consistent with previous research (i.e., [58]), highlighting the significant impact of academic challenges on mental health, which collectively contribute to a general dimension that we have termed *psychological well-being disturbances*. This outcome is underscored by the PCA analysis, which demonstrated that a single component accounted for 58.5% of the total variance, indicating that these psychological factors are not only correlated but also share a common underlying dimension. Furthermore, the high correlation between prospective and retrospective memory complaints suggests that subjective memory impairment plays a critical role in student psychological well-being and cognitive dimensions.

The anti-correlated relationship observed between internal and external LOC behaviors highlights the complexity and the significant insights into how individuals perceive their ability to influence life outcomes. This finding suggests that as individuals develop a stronger internal LOC, they may simultaneously diminish reliance on external factors to explain their experiences. This dynamic is crucial in psychology, particularly in understanding coping mechanisms and resilience among students facing academic and personal challenges.

The analysis of the Big Five personality traits revealed nuanced relationships with PWBD, SMC, and LOC, highlighting the importance of considering individual differences in shaping psychological outcomes. Extraversion and conscientiousness were positively associated with higher levels of internal control and mindfulness, suggesting their protective role against psychological distress. Conversely, neuroticism was linked to increased external control and memory complaints, corroborating its role as a vulnerability factor for mental health issues. Openness showed a strong positive loading on internal LOC, suggesting that individuals with a higher level of openness are more likely to believe they can influence their circumstances through their actions. Individuals with a higher level of openness may engage more actively in seeking solutions and coping strategies, which may contribute to better mental health outcomes and reduce PWBD. Conscientiousness demonstrated a positive correlation with both internal LOC and mindfulness, implying that (i) individuals with a higher level of conscientiousness tend to be more organized, responsible, and goal-oriented, enhancing their ability to control outcomes; (ii) conscientious students might also adopt more effective stress management strategies, thereby lowering anxiety and depression risks.

Collectively, the findings support theoretical models suggesting that personality traits influence mental health through pathways like stress exposure, stress appraisal, and coping strategies. The mediating roles of stress mindset and coping flexibility, in the relations between the Big Five personality traits and psychological distress, as demonstrated in previous research [59], further elucidate these complex processes. However, the study suffered from several limitations. One significant limitation was the gender imbalance, with only nine male participants, which restricts the generalizability of the results due to potential gender-specific differences in the observed outcomes. Additionally, the cross-sectional nature of the study, along with the use of self-report measures, introduces potential biases that may affect the accuracy of the findings and restrict the interpretation of relationships between the variables. Regarding the sampling procedure, participants were selected using a convenience sampling method from a pool of undergraduate students. Although this method was used, we acknowledge its limitations, and the findings should be considered preliminary insights rather than generalizations to a broader population.

## 5. Conclusions

In conclusion, this study underscores the importance of personality traits and psychological factors in academic settings. The PWBD dimension offers a valuable dimension for future research and interventions to enhance student well-being.

Educational institutions should incorporate personality-based approaches such as tailored mindfulness programs or stress-management workshops to support students with traits like neuroticism in coping with academic challenges. Additionally, implementing screening tools to identify at-risk students early could enable the development of targeted support interventions.

Future studies should replicate these findings in broader populations and assess targeted interventions based on personality traits.

## Figures and Tables

**Figure 1 ijerph-22-00132-f001:**
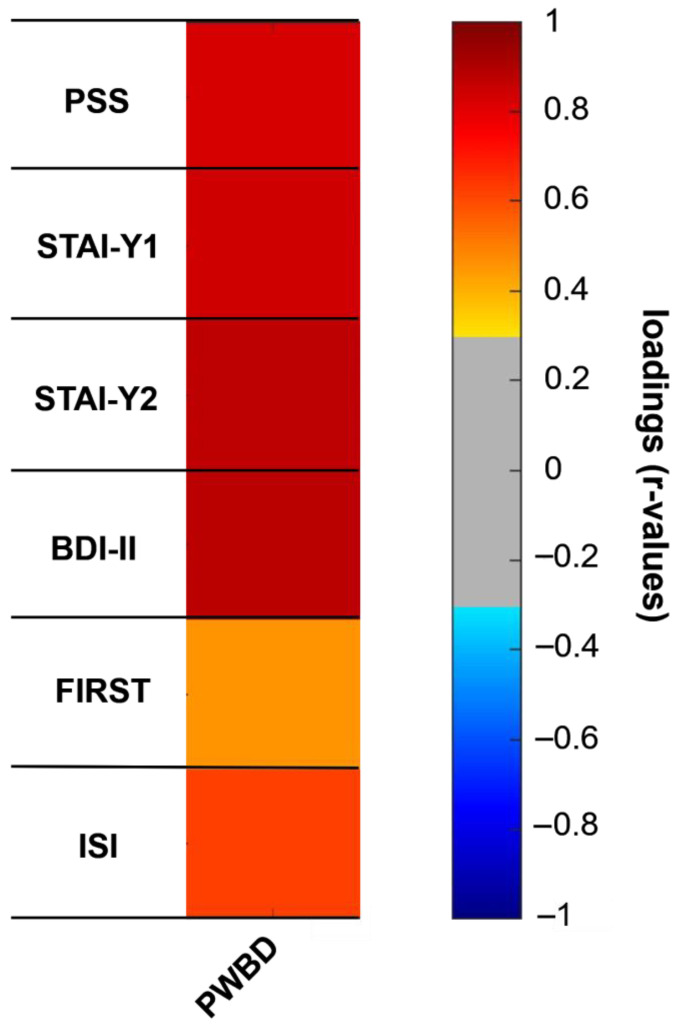
In the present figure, the loadings of the considered variables on the extracted PCA component (PWBD) are presented. Significant positive loadings are identified by red tones (orange to dark red). The threshold for significance at *p* < 0.05 is |rth| = 0.31.

**Figure 2 ijerph-22-00132-f002:**
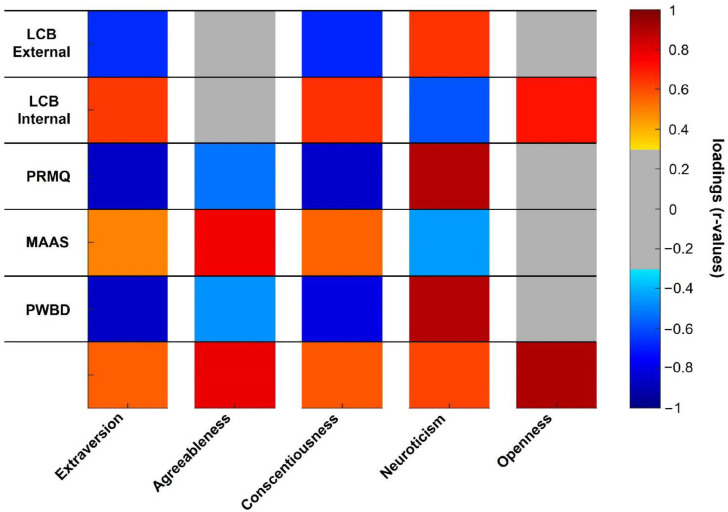
For each PCA (i.e., extraversion, agreeableness, conscientiousness, neuroticism, and openness), the variables’ loadings on the PC component on which the trait to be investigated had a significant loading are reported. Note that the loadings of Big Five Inventory traits are presented on the lower row. Significant positive loadings are identified by red tones (orange to dark red), whereas significant negative ones by blue tones (light to dark blue). Non-significant loadings are colored in gray.

**Table 1 ijerph-22-00132-t001:** Descriptive statistics (mean, standard deviation (SD), median, and interquartile range (IQR)) are reported for each variable collected. For each psychometric scale, the Cronbach’s alpha measure of internal consistency is presented.

	Mean	SD	Median	IQR	Cronbach’s Alpha
Age	19.45	1.62	19	0.75	---
Education	13.04	0.37	13	0	---
LCB-E	19.97	5.12	20	6	0.703
LCB-I	23.58	4.23	24	6	0.681
LCB	31.39	7.91	30	12.75	0.724
BDI-II	17.91	9.75	18	15.25	0.882
STAY-Y1	45.12	11.45	44	16.50	0.916
STAY-Y2	50.87	8.37	52	9.75	0.835
BFI-E	19.46	6.37	19	9	0.849
BFI-A	27.64	5.63	28	8	0.711
BFI-C	24.31	6.48	24	7.75	0.812
BFI-N	23.69	4.21	25	5.75	0.709
BFI-O	35.67	6.06	35	9.75	0.774
PSS	26.81	6.11	26	7.50	0.804
PRMQ P	23.13	5.77	22	8.50	0.828
PRMQ R	19.33	4.10	20	6.75	0.746
PRMQ	42.46	8.93	43	10	0.846
MAAS	3.87	0.90	4	1.32	0.878
FIRST	22.52	5.96	22	8	0.849
ISI	8.43	4.97	8	7.50	0.778

**Table 2 ijerph-22-00132-t002:** The loadings of the considered psychometric variables on the extracted component, along with their significance, are presented. Note that thresholds for significance at *p* = 0.05 and *p* = 0.001 are r = 0.31 and r = 0.43, respectively.

Variables	Loadings	*p*-Values
PSS	0.827	0.001
STAI-Y1	0.836	0.001
STAI-Y2	0.870	0.001
BDI-II	0.882	0.001
FIRST	0.460	0.001
ISI	0.619	0.001

**Table 3 ijerph-22-00132-t003:** The loadings of the considered psychometric variables on the extracted component, along with their significance, are presented. Note that thresholds for significance at *p* = 0.05 and *p* = 0.001 are r = 0.31 and r = 0.45, respectively.

Variables	Loadings	*p*-Values
BFI-Extraversion	0.556	0.001
LCB-External	−0.668	0.001
LCB-Internal	0.637	0.001
PRMQ-Total	−0.862	0.001
MAAS	0.489	0.001
PWBD	−0.862	0.001

**Table 4 ijerph-22-00132-t004:** The loadings of the considered psychometric variables on the PCA component related to agreeableness, along with their significance, are presented. Note that thresholds for significance at *p* = 0.05 and *p* = 0.001 are r = 0.31 and r = 0.44, respectively.

Variables	Loadings	*p*-Values
BFI-Agreeableness	0.785	0.001
LCB-External	−0.083	0.972
LCB-Internal	−0.063	0.993
PRMQ-Total	−0.524	0.001
MAAS	0.767	0.001
PWBD	−0.469	0.001

**Table 5 ijerph-22-00132-t005:** The loadings of the considered psychometric variables on the extracted component, along with their significance, are presented. Note that thresholds for significance at *p* = 0.05 and *p* = 0.001 are r = 0.32 and r = 0.45, respectively.

Variables	Loadings	*p*-Values
BFI-Conscientiousness	0.571	0.001
LCB-External	−0.680	0.001
LCB-Internal	0.656	0.001
PRMQ-Total	−0.853	0.001
MAAS	0.547	0.001
PWBD	−0.807	0.001

**Table 6 ijerph-22-00132-t006:** The loadings of the considered psychometric variables on the extracted component, along with their significance, are presented. Note that thresholds for significance at *p* = 0.05 and *p* = 0.001 are r = 0.31 and r = 0.44, respectively.

Variables	Loadings	*p*-Values
BFI-Neuroticism	0.614	0.001
LCB-External	0.644	0.001
LCB-Internal	−0.590	0.001
PRMQ-Total	0.893	0.001
MAAS	−0.450	0.001
PWBD	0.894	0.001

**Table 7 ijerph-22-00132-t007:** The loadings of the considered psychometric variables on the PCA component related to openness, along with their significance, are presented. Note that thresholds for significance at *p* = 0.05 and *p* = 0.001 are r = 0.32 and r = 0.46.

Variables	Loadings	*p*-Values
BFI-Openness	0.902	0.001
LCB-External	−0.243	0.221
LCB-Internal	0.707	0.993
PRMQ-Total	−0.136	0.792
MAAS	0.189	0.481
PWBD	−0.034	1.000

## Data Availability

The data presented in this study are available upon reasonable request from the corresponding author.

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
