# Peer review of "The Interplay of Personality Traits and Psychological Well-Being: Insights from a Study of Italian Undergraduates"

_ijerph, 2025, doi:10.3390/ijerph22020132_

Round 1

Reviewer 1 Report

Comments and Suggestions for Authors

The article focuses on a highly relevant issue related to student well-being. This exploratory study of 67 students examines the link between personality traits, internal vs. external control, and behaviors related to well-being (anxiety, depression, sleep, difficulty memorizing). I would like to thank the authors for reviving the locus of control, which has been little studied in the early 21st century.

The review of the literature presented is relevant to the authors' initial question; it enables us to follow the authors' path to determine their problematic and then to choose the variables taken into consideration.

My comments mainly concern the experimental part. The section on participants and procedure is not detailed enough: how were the volunteers recruited? What are the characteristics of these students (we learn in the results section about the distribution of the population; is it relevant to keep the 8 men?)? Is the number of participants sufficient for what the authors wish to demonstrate and for the number of variables taken into consideration (calculation of g-power)?

In the presentation of the various measures, the article would gain in precision by giving additional indications on the psychometric qualities and the relevance of the tool with regard to the population ; likewise, examples of items for each of the tools' dimensions.

The results are clearly presented. The only comment concerns Figure 1, which we feel is essential for understanding. Figure 2, on the other hand, is very telling.

Form: APA standards not always respected in the presentation of results

Author Response

We sincerely thank you for the feedback provided on our manuscript. We greatly appreciate the time and effort dedicated to evaluating our work, as well as the valuable suggestions provided to improve its quality and clarity.

We have carefully addressed each comment raised by the reviewers. Wherever possible, we have incorporated their suggestions into the revised manuscript to enhance its scientific rigor and overall presentation. For comments that required further clarification or a different approach, we have provided detailed justifications in our responses.

Below, we present a point-by-point reply to each comment. Reviewers’ comments are reproduced in italicized text for clarity, followed by our responses and any relevant changes made to the manuscript.

Thank you once again for your guidance and support throughout this process.

Sincerely,

Dr. Graziella Orrù

Reviewer 1:

  1. My comments mainly concern the experimental part. The section on participants and procedure is not detailed enough: how were the volunteers recruited? What are the characteristics of these students (we learn in the results section about the distribution of the population; is it relevant to keep the 8 men?)? Is the number of participants sufficient for what the authors wish to demonstrate and for the number of variables taken into consideration (calculation of g-power)?

Authors:

We thank the reviewer for raising this point. We choose to maintain 8 men in the students’ sample as they showed no significant difference as compared to females either in demographic or psychometric variables, as verified using Independent Samples T-tests with unequal variance assumed (this assumption is motivated by the large difference in sample size between men and females). In the table below we report, for each variable of interest, the statistics related to the comparison (t-value, p-value and p-value after Benjamini-Hochberg correction for multiple comparison, pBH, Benjamini & Hochberg, 1995).

t-value

p-value

pBH

AGE

-1.59

0.13

0.43

EDUCATION

1.00

0.35

0.50

LCB_E

-1.12

0.29

0.50

LCB_I

0.07

0.95

0.95

LCB

-0.74

0.48

0.60

BDI II

-1.56

0.15

0.43

STAI_Y1

-1.90

0.09

0.43

STAI_Y2

-1.12

0.30

0.50

BFI_E

1.67

0.13

0.43

BFI_A

0.28

0.79

0.89

BFI_C

-0.83

0.43

0.58

BFI_N

-1.18

0.27

0.50

BFI_O

1.86

0.10

0.43

PSS

-1.51

0.17

0.43

PRMQ_P

1.15

0.28

0.50

PRMQ_R

0.08

0.94

0.95

PRMQ

-0.99

0.35

0.50

MAAS

-0.26

0.80

0.89

FIRST

-1.77

0.11

0.43

ISI

-1.70

0.12

0.43

Reference

Benjamini, Y., & Hochberg, Y. Controlling the false discovery rate: a practical and powerful approach to multiple testing. J. Roy. Stat. Soc. B Met., 1995, 57, 289-300.

Additionally, in regard to the question:  is the number of participants sufficient for what the authors wish to demonstrate and for the number of variables taken into consideration (calculation of g-power)? We thank the reviewer for raising this fundamental issue. The study is based on principal component analysis with the aim of identifying patterns in the correlations between variables. These patterns are in turn used to infer the existence of underlying latent variables (i.e. factors) in the data. In this context a number of studies (Barrett & Kline, 1981; Gorsuch, 1983; de Winter et al., 2009), have demonstrated that the reliability of a PCA analysis (and hence its statistical power), stands on two sufficient conditions:

  1. A sample size (i.e. number of observations/participants) higher than 50.
  2. A sample size to variable ratio ≥ 5.

Each PCA conducted in our study satisfied the two conditions: the sample size was 67 and the sample size to variable ratio > 10 (6 variables for each PCA).

We have added these considerations at pages 4 and 5 (lines 195-209), in the Methods and Measures section (Principal Components Analysis subsection), we cite for your convenience:

“The reliability (and hence the statistical power) of a PCA, stands on two sufficient conditions (see Barrett & Kline, 1981; Gorsuch, 1983; de Winter et al., 2009): 

  • A Sample Size (i.e. number of observations/participants) higher than 50.
  • A Sample Size to Variable ratio ≥ 5.

Throughout the years, the determination of the minimum sample size and of the sample size to variable ratio have been a matter of ongoing debate among researchers: it is thus fair to underline that, as demonstrated by de Winter and colleagues (2009), when data are well conditioned (high loadings, low number of factors, relatively high number of variables), the two conditions can be relaxed (e.g. a PCA can yield reliable results also for sample sizes well below 50).

However, to ensure the reliability of our findings we adopted the two thresholds mentioned at the beginning of the paragraph. Indeed, each PCA conducted in the study satisfied the two conditions: the sample size was always 67 and the sample size to variable ratio higher than 10 (6 variables for each PCA), and as such well above the lower limit of 5.”

  1. In the presentation of the various measures, the article would gain in precision by giving additional indications on the psychometric qualities and the relevance of the tool with regard to the population; likewise, examples of items for each of the tools' dimensions.

Thank you for the suggestion. As noted in the Psychometric Measures paragraph (pages 3 and 4), we have implemented the indication required.

  1. The results are clearly presented. The only comment concerns Figure 1, which we feel is essential for understanding. Figure 2, on the other hand, is very telling.

We thank the reviewer for pointing out the issue. We have now replaced Figure 1 with a new one, using the same style as Figure 2. This change is visible on page 10 of the manuscript.

  1. Form: APA standards not always respected in the presentation of results

We sincerely thank the reviewer for the feedback. We have modified the reference within the manuscript according to the journal's guidelines, which require references to be cited using numbered square brackets (e.g., [1], [1–3], or [1,3]). Additionally, within the guidelines section is reported the following statement: “Your references may be in any style, provided that you use the consistent formatting throughout. It is essential to include author(s) name(s), journal or book title, article or chapter title (where required), year of publication, volume and issue (where appropriate) and pagination. DOI numbers (Digital Object Identifier) are not mandatory but highly encouraged. The bibliography software package EndNote, Zotero, Mendeley, Reference Manager are recommended.”. Therefore, it appears that it is not required necessarily the APA style. In order to make the reference we have used Zotero.  

Link: https://www.mdpi.com/journal/ijerph/instructions (Paragraph: Free Format Submission).

Please, find attached the revised version of the Manuscript

Reviewer 2 Report

Comments and Suggestions for Authors

Dear Authors,

I have read your manuscript with interest; it deals with a crucial and actual topic. Despite your effort in data analysis, more significant changes are required in the manuscript's structure to improve its quality before publication. Following are my comments and suggestions:

1. Line 67-68: you mention data on stress, anxiety, and depression in university students. Maybe more updated research could be more representative of the current mental health conditions in the student population.

2. Please add a paragraph outlining the study's aims and hypothesis at the end of the introduction. 

3. Participants and procedure:

More information is needed: specify the sampling procedure and the adequacy of the sample size in relation to the analysis; please conduct an a priori analysis to determine the necessary sample size to justify your sample that seems small in size for a cross-sectional study. 

Moreover, the characteristics of the participants should be moved from the Results section to the Participants section. Information on the degree course attended and the type of study are necessary.  

Please specify the online platform used for data detection and add the instructions included in the survey.

4. Measures:

Add one sample item and the Cronbach's alpha for each scale, calculated on the sample of the study

5. Paragraph 2.3. should be renamed as 'Data analysis', comprising 2.3.1., 2.3.2., 2.4.

6. Paragraph 2.5. should be renamed as 'Results', comprising and merging 2.5.1., 2.5.2., 2.5.3., 2.5.4. 

7. Paragraph 3 will become 'Descriptives'

8. A Limitation section must precede the Conclusions, highlighting the study's limits (some of them are, i.e., the cross-sectional nature of the study and the exclusive use of self-report measures); moreover, suggestions for practical implications should be added in the Conclusions paragraph.

Author Response

Reviewer 2

  1. Line 67-68: you mention data on stress, anxiety, and depression in university students. Maybe more updated research could be more representative of the current mental health conditions in the student population.

Thank you.  We agree that more recent research would be more representative and appropriate. As a result, we have replaced the previous study with a more up-to-date one, as shown in line 68 at page 2.

  1. Please add a paragraph outlining the study's aims and hypothesis at the end of the introduction. 

We thank the reviewer for the feedback provided. We have now expanded our aims and hypotheses (lines 91-102, pages: 2 and 3), as stated below:

The primary aim of this study was to investigate how personality traits, specifically the Big Five dimensions, interacted with locus of control (LOC) to influence various psychological outcomes among first-year university students. We hypothesized that personality traits, particularly Extraversion, Conscientiousness, and Neuroticism, would have a significant impact on psychological well-being, with internal LOC potentially serving as a moderating factor. Specifically, we expected that students with higher levels of Extraversion and Conscientiousness would experience better psychological outcomes, while those exhibiting higher Neuroticism and external LOC would report more negative psychological effects, including increased levels of anxiety, depression, and sleep disturbances. This research aimed to provide insights that could guide the development of targeted interventions to support the mental health and well-being of university students, particularly in their first year, and enhance their academic success.”

  1. Participants and procedure:

More information is needed: specify the sampling procedure and the adequacy of the sample size in relation to the analysis; please conduct an a priori analysis to determine the necessary sample size to justify your sample that seems small in size for a cross-sectional study. 

We thank the reviewer for raising the issue. We must first underline that the adequacy of a sample size strongly depends on the methodology of analysis which is chosen based on the study aims. Our study was conceived to:

  • Verify the existence of a common latent factor underlying stress, anxiety, depression and insomnia.
  • Highlight the relationships (if any), between each Big-Five Inventory personality trait and psychological well-being domains.

On the basis of these aims, we used a Principal Component Analysis approach. The reliability of a PCA (and hence its statistical power), stands on two sufficient conditions (see Barrett & Kline, 1981; Gorsuch, 1983; de Winter et al., 2009):

  • A sample size (i.e. number of observations/participants) higher than 50.
  • A sample size to Variable ratio ≥ 5.

Each PCA conducted in our study satisfied the two conditions: the sample size was 67 and the sample size to variable ratio > 10 (6 variables for each PCA), and as such well above the lower limit of 5.

We have added these considerations in the Methods and Measures section (Principal Components Analysis subsection), we cite for your convenience:

The reliability (and hence the statistical power) of a PCA, stands on two sufficient conditions (see Barrett & Kline, 1981; Gorsuch, 1983; de Winter et al., 2009): 

  • A Sample Size (i.e. number of observations/participants) higher than 50.
  • A Sample Size to Variable ratio ≥ 5.

Throughout the years, the determination of the minimum sample size and of the sample size to variable ratio have been a matter of ongoing debate among researchers: it is thus fair to underline that, as demonstrated by de Winter and colleagues (2009), when data are well conditioned (high loadings, low number of factors, relatively high number of variables), the two conditions can be relaxed (e.g. a PCA can yield reliable results also for sample sizes well below 50).

However, to ensure the reliability of our findings we adopted the two thresholds mentioned at the beginning of the paragraph. Indeed, each PCA conducted in the study satisfied the two conditions: the sample size was always 67 and the sample size to variable ratio higher than 10 (6 variables for each PCA), and as such well above the lower limit of 5.”

Regarding the sampling procedure, participants were selected using a convenience sampling method from a pool of undergraduate students. Although a convenience sampling method was used, we acknowledge its limitations and emphasize that the findings are intended to provide preliminary insights rather than generalizations to a broader population. We have added this limitation at page 15, lines 497-501.

Moreover, the characteristics of the participants should be moved from the Results section to the Participants section. Information on the degree course attended and the type of study are necessary.  

Thank you for your suggestion, we removed the characteristics from the Results.

Please specify the online platform used for data detection and add the instructions included in the survey.

Thank you. We implement the manuscript with the required information (page 3; lines: 115-124), as stated below:

“Data collection was conducted via an online survey platform (Google Form). Participants were informed that the purpose of the research was to psychometrically assess personality domains and their association with indicators of mental well-being through the administration of specific questionnaires. General instructions provided to the participants in the survey were as follows: 'We are conducting research aimed at psychometrically assessing personality domains and their association with indicators of mental well-being by administering specific questionnaires/tests. Completing the questionnaire will take approximately 15 minutes.' For each individual questionnaire, participants were provided with the specific instructions relevant to that questionnaire, which remained consistent with the original instructions for each test.”

  1. Measures: Add one sample item and the Cronbach's alpha for each scale, calculated on the sample of the study

We thank the reviewer for the suggestion: we added the Cronbach’s alpha for each scale. The information has been added to Table 1 and the manuscript has been modified accordingly. The rest of information required has been added in the measures section.

We cite for your convenience:

In 2.3.1 Statistical Methods Overview: “The reliability of each psychometric scale was assessed by calculating its Cronbach’s alpha value.”

Table 1 is now as follows:

Table 1. Descriptive statistics (mean, standard deviation, SD, median and interquartile range, IQR) are reported for each variable collected. For each psychometric scale the Cronbach’s alpha measure of internal consistency is presented.

Mean

SD

Median

IQR

Cronbach’s alpha

Age

19.45

1.62

19

0.75

---

Education

13.04

0.37

13

0

---

LCB-E

19.97

5.12

20

6

0.703

LCB-I

23.58

4.23

24

6

0.681

LCB

31.39

7.91

30

12.75

0.724

BDI-II

17.91

9.75

18

15.25

0.882

STAY-Y1

45.12

11.45

44

16.50

0.916

STAY-Y2

50.87

8.37

52

9.75

0.835

BFI-E

19.46

6.37

19

9

0.849

BFI-A

27.64

5.63

28

8

0.711

BFI-C

24.31

6.48

24

7.75

0.812

BFI-N

23.69

4.21

25

5.75

0.709

BFI-O

35.67

6.06

35

9.75

0.774

PSS

26.81

6.11

26

7.50

0.804

PRMQ P

23.13

5.77

22

8.50

0.828

PRMQ R

19.33

4.10

20

6.75

0.746

PRMQ

42.46

8.93

43

10

0.846

MAAS

3.87

0.90

4

1.32

0.878

FIRST

22.52

5.96

22

8

0.849

ISI

8.43

4.97

8

7.50

0.778

From the Descriptives subsection: …” All psychometric scales showed satisfactory internal consistency levels (see Cronbach’s Alpha, Table 1): all the scales except for LCB-I (0.681), had Alpha values higher than 0.7, although values higher than 0.6 are still considered adequate (Taber, 2018).”

  1. Paragraph 2.3. should be renamed as 'Data analysis', comprising 2.3.1., 2.3.2., 2.4.

Thank you, we have made the changes as suggested and adjusted the paragraph numbering accordingly.

  1. Paragraph 2.5. should be renamed as 'Results', comprising and merging 2.5.1., 2.5.2., 2.5.3., 2.5.4. 

Thank you. We understand the reviewer’s concern. As a matter of fact, these paragraphs do not significantly contribute to the Methods section, but they rather anticipate findings that are already well described in the results section. Based on these considerations we have decided to delete them. 

  1. Paragraph 3 will become 'Descriptives'

Thank you, we have made the changes as suggested

  1. A Limitation section must precede the Conclusions, highlighting the study's limits (some of them are, i.e., the cross-sectional nature of the study and the exclusive use of self-report measures); moreover, suggestions for practical implications should be added in the Conclusions paragraph.

Thank you for your suggestion. All the changes are visible at page 15 (lines: 491-501).

Please find attached the revised version of the manuscript. 

Round 2

Reviewer 2 Report

Comments and Suggestions for Authors

Thank you for having revised the manuscript according to the comments